# Protective Effects of ALDH1A Enzyme Inhibition on *Helicobacter*-Induced Colitis in Smad3^−/−^ Mice are Associated with Altered α4ß7 Integrin Expression on Activated T Cells

**DOI:** 10.3390/nu12102927

**Published:** 2020-09-24

**Authors:** Audrey Seamons, Michael Haenisch, Stacey Meeker, Olesya Pershutkina, Thea Brabb, Piper M. Treuting, Jisun Paik

**Affiliations:** Department of Comparative Medicine, University of Washington, Seattle, WA 98195, USA; auds@uw.edu (A.S.); haenisch@uw.edu (M.H.); meeker.79@osu.edu (S.M.); olesya@uw.edu (O.P.); thea@uw.edu (T.B.); treuting@uw.edu (P.M.T.)

**Keywords:** retinoic acid, ALDH1A enzymes, immunity, inflammatory bowel disease, mouse model

## Abstract

Many inflammatory bowel disease (IBD) patients require surgical intervention due to limited pharmacological treatment options. Antibodies targeting α4ß7, a gut-homing integrin, are one of the most promising IBD treatments. As retinoic acid (RA) regulates expression of gut-homing proteins including α4ß7 integrin, we tested if ALDH1A enzymes in the RA synthesis pathway could be targeted for IBD treatment using a potent inhibitor, WIN 18,446. Age- and sex-matched *Smad3*^−/−^ mice were fed a diet with and without WIN 18,446 for 3 weeks before triggering inflammation with *Helicobacter bilis* infection. Colitis was evaluated by histopathology one week following the IBD trigger, and T cell subsets were evaluated before and after the IBD trigger. WIN 18,446 treatment significantly reduced IBD severity in *Smad3*^−/−^ mice and reduced expression of α4ß7 integrin on multiple activated CD4^+^ T cell subsets. This change was associated with increased ratios of induced regulatory T cells to Th17 cells during the inflammatory response in the draining lymph nodes. These studies indicate that RA reduction via ALDH1A enzyme inhibition is a potential new target for IBD treatment. Further studies are needed to examine its effects on other types of immune cells, to evaluate the efficacy window for this target, and to determine its efficacy in other animal models of IBD.

## 1. Introduction

As the precise etiology of human inflammatory bowel diseases (IBDs) is not known, treatment for these diseases has focused on symptom relief, using anti-inflammatory drugs such as corticosteroids and 5-ASA or general immunomodulatory drugs such as thiopurines, and methotrexate [1]. In comparison to the general immune modulators, newer monoclonal antibody-based drugs target specific immune modulators or pathways such as TNFα, IL12/23, IL23, integrins and integrin receptors, aiming at suppressing inflammation with improved efficacy [1,2,3,4,5]. However, not a single treatment option has shown efficacy in all IBD cases, achieving clinical remission, likely due to diverse underlying etiologies for IBD. While monoclonal antibody treatments are promising, many patients do not respond to the therapy initially or lose responsiveness to the therapy with time [6]. Anti-drug antibody development and diminished capacity to fight infection are other concerns with antibody-based biologicals [2,3]. Additionally, biologicals are given by infusion and/or injection, requiring patients to visit a clinic frequently and, therefore, is a significant burden to them. Despite availability of these treatment options, IBD-related surgical intervention occurs for a large percentage of IBD patients—in Sweden, one study showed that 27% of ulcerative colitis patients and 51% of Crohn’s disease patients eventually needed surgery [7]. Thus, new targets and drugs with more convenient delivery methods (e.g., oral) will greatly improve current treatment options for IBD. To address this need, we examined whether the retinoic acid (RA) synthesis pathway can be a potential new drug target for IBD treatment.

RA, a biologically active metabolite of vitamin A, is involved in various functions in the body, including immunity [8]. RA is synthesized by a two-step oxidation process from retinol [9,10]. The first reversible step produces retinal from retinol by two families of enzymes, alcohol dehydrogenases and short-chain dehydrogenase/reductases. The second irreversible step is catalyzed by a family of aldehyde dehydrogenases, ALDH1A enzymes, ALDH1A1, ALDH1A2 and ALDH1A3. In the gut, ALDH1A1 and ALDH1A2 appear to be the major producers of RA. ALDH1A1 is expressed in epithelial cells while ALDH1A1 and ALDH1A2 are expressed in dendritic cells (DCs) and macrophages in the lamina propria, mesenteric lymph node (MLN) and Peyer’s patches [10]. MLN stromal cells express all three isoforms of ALDH1A enzymes [11]. RA is critically involved in gut immunity by maintaining the integrity of the gut barrier through effects on epithelial cells [12,13,14], influencing T cell differentiation [15,16] and IgA production [15], and imprinting T and B cells with gut-homing ability [15,17]. Because RA can enhance both inflammatory and regulatory immune responses depending on its concentration and the context of other immune modulatory molecules [15,16,18], it is unclear how changes in RA signaling influence inflammatory diseases such as IBD. In agreement with the known beneficial roles of RA in regulatory immunity, RA treatment has been shown to suppress IBD in the adoptive transfer, TNF^ΔARE^ (ileitis) and chemically induced murine models of IBD [19,20,21,22,23]. On the other hand, it has been shown that RA can induce Th17 immunity [18], which is associated with human IBDs, in the context of IL-15 and IL-6 in the intestinal mucosa and thus, it may not be beneficial as a treatment of intestinal inflammation associated with increased levels of IL-15 such as in IBD [24]. Indeed, it has been demonstrated that RA treatment is associated with more severe IBD due to decreased mucin expression in a zebrafish model of chemically induced colitis [25]. In humans, there is controversy on whether RA treatment for acne is associated with increased risk for development of IBD [26,27,28]. In cell culture studies, Sanders et al. [29] showed that ALDH1A activities are increased in antigen presenting cells from patients with Crohn’s disease. Interestingly, Kang et al. demonstrated that both high and low vitamin A conditions ameliorate gut inflammation through induction of distinct populations of regulatory T cells (Tregs) [30]. Given these opposing observations, we used a mouse model of IBD, *Smad3*^−/−^ mice, to test whether targeting the RA synthesis pathway would impact disease development.

*Smad3*^−/−^ mice are deficient in TGFß signaling, a pathway that promotes both regulatory immune responses (induction of Tregs) and inflammatory immune responses (Th17 cell differentiation) [31] and is critical to tissue-repair responses [32]. SMAD3 is a transcription factor downstream of TGFß, and mice lacking SMAD3 are prone to inflammation [33,34] and develop IBD with either bacterial [33] or chemical triggers [35]. The clinical manifestation of acute IBD based on diarrhea, weight loss and/or anal inflammation in *Smad3*^−/−^ mice is usually short-lived (the duration varies in our laboratory and typically peaks at 5–7 days post *H. bilis* infection). However, the inflammation phase characterized by mucosal hyperplasia and chronic inflammatory infiltrates has been noted even at 5 weeks post infection with *H. bilis* by histological analysis [33]. Thus, chronic inflammation occurs in this mouse model following *H. bilis* infection that is not observable by clinical symptoms. The role that T cells play in IBD of *Smad3*^−/−^ mice is not well defined, though others have observed increased CD4^+^ and CD8^+^ activated T cells in MLN (in addition to increased NK cells) and increased CD3^+^ cells and Granzyme B^+^ cells in the proximal colon following *H. hepaticus* infection in *Smad3*^−/−^ mice [36]. Peak clinical disease in *H. bilis*-infected *Smad3*^−/−^ mice is characterized by significantly increased T cells and macrophages based on CD3 and F4/80 staining, respectively, with upregulation of MHC Class II as shown by immunohistochemistry of colon tissue [37]. The influx of T cells into the colon during disease implicates them in the disease process in this model.

To target the RA synthesis pathway, we used a specific inhibitor of ALDH1A enzymes, WIN 18,446 [38,39,40]. WIN 18,446 is a proven strong inhibitor of ALDH1A1 and ALDH1A2, reducing RA in vitro and in vivo [38,39,40]. Thus, it is a useful tool to determine how RA inhibition affects gut inflammation in vivo.

## 2. Materials and Methods

### 2.1. Smad3^−/−^ Mice and Diets

Male and female *Smad3*^−/−^ mice (129-*Smad3*^tm1Par^/J) were bred in our facility by both homozygous and heterozygous matings. Mice were housed in individually ventilated cages (Allentown, Allentown, NJ, USA) with autoclaved corncob bedding (Andersons, Maumee, OH, USA) and provided with irradiated rodent chow (#5053, Lab Diet, St. Louis, MO, USA) and autoclaved, acidified (pH 2.4–2.8) water. During experiments, mice were given either purified rodent diet (AIN93M, TestDiet, St. Louis, MO, USA) or AIN93M containing 2 mg WIN 18,446/g diet [38]. Mice were housed in a room excluding *Helicobacter* in a specific pathogen-free facility, and mice experimentally infected with *Helicobacter* were kept on a separate side of the rack from uninfected mice with a separate hood/supplies used for husbandry. All protocols for animal studies were approved by the Institutional Animal Care and Use Committee at the University of Washington.

### 2.2. Study to Test Effects of WIN 18,446 in a Mouse Model of IBD

Age- and sex-matched *Smad3*^−/−^ mice were fed either AIN93M diet with or without WIN 18,446 (Acros Organics, Geel, Belgium) for 3 weeks, weighed weekly and observed for general health. All mice were then infected with *H. bilis* by oral gavage [37], and monitored for IBD symptoms, including declining body condition, weight loss and diarrhea. Mice were euthanized by CO_2_ asphyxiation and cardiocentesis if they met predetermined criteria (>20% loss of body weight, body condition score ≤ 2, dehydration) or the study end point (one-week post *H. bilis* infection). At necropsy, colon, cecum, and MLN were fixed in 10% neutral buffered formalin and processed as previously reported for histological scoring with minor modifications by a board-certified veterinary pathologist (PMT) blinded to the experimental groups [37,41]. Briefly, inflammation was scored on three sections of colon (proximal, mid and distal) and cecum based on severity of mucosal epithelial changes, degree of inflammation and extent of pathology without weighting. The maximum score of each region is 16 and thus, the total possible maximum score is 64.

### 2.3. Studies to Characterize Immune Cells in Response to RA Reduction

To determine effects of RA reduction by ALDH1A enzyme inhibition on immune cell functions, we performed two studies. In the first study, immune cell changes in *Smad3*^−/−^ mice were evaluated in response to WIN 18,446 treatment during steady state. Age- and sex-matched *Smad3*^−/−^ mice were fed AIN93M diet with or without WIN 18,446 for 3 weeks and immune cells in MLN and peripheral (inguinal, axial, and brachial) lymph nodes were evaluated. In the second study, immune cells were characterized during inflammation in *Smad3*^−/−^ mice fed a diet with or without WIN 18,446. Age- and sex-matched *Smad3*^−/−^ mice were fed AIN93M diet with or without WIN 18,446 for three weeks and then infected with *H. bilis*. Four days post *H. bilis* infection, all mice were euthanized, and immune cells were isolated from MLN and lamina propria of cecum (half) and colon.

### 2.4. Immune Cell Characterization

Single-cell suspensions were generated from MLN, peripheral lymph nodes (PLN) or lamina propria harvested from ½ cecum and 4 cm proximal colon [37]. Leukocyte-sized cells were counted using a hemocytometer after trypan-blue staining. The antibodies used in the studies (steady state vs. during inflammation) were slightly different from each other, and staining combinations are indicated in Table 1. All samples were incubated with purified anti-mouse CD16/CD32 (Mouse BD FcBlock™, BD Biosciences) 10 min prior to and throughout the surface-staining antibody protocol. MLN and PLN cells (1 × 10^6^ cells per stain) were stained in the steady state study, and MLN cells (1 × 10^6^ cells per stain) and lamina propria leukocytes (LPL, 5 × 10^5^ cells per stain) were stained in the inflammation study. Dead cells were excluded in LPL preparations using the LIVE/DEAD blue kit (eBioscience/Thermofisher). Cells were singly stained for all markers and single stains used to make initial cytometer settings. No-stain controls for each cell type were also generated. For α4β7-PE, a substitute antibody that can separate positive and negative cell populations more clearly was also used. In the inflammation studies, fluorescence minus one (F-1) controls were also generated for antibodies that stain rare populations or for antibodies for which staining levels do not form well-separated populations (e.g., antibodies staining cytokines or α4β7 integrin). Expression of effector cytokines on CD4^+^ T cells (IL-17 and IFNγ) was determined by stimulating cells in vitro for 4 h at 37 °C in RPMI containing 10% FBS, 100 units/mL Penicillin/Streptomycin, 100 ng/mL Phorbol 12-myristate 13-acetate (PMA, Sigma-Aldrich), 1µg/mL Ionomycin (EMD Millipore), and 1:1000 dilution of Golgi Plug (contains Brefeldin A, BD Bioscience, San Jose, CA, USA). Unstimulated cells incubated in media without PMA and Ionomycin were stained and used as control samples. Intracellular staining was performed using the BD Biosciences Intracellular Cytokine Staining kit for IL-17 and IFNγ or the Foxp3 eBioscience staining kit for Foxp3 and RORγt. Stained cells were evaluated on an LSRII (BD Biosciences). Between 50,000 and 600,000 events were collected through live and singles gates depending on the stain and tissue type. Data were analyzed using FlowJo v.10 software. Gating strategies for select populations from each stain (Stains 1–3 as described in Table 1) are shown in Appendix A.

### 2.5. Enzyme Activity Assay for RA Synthesis in Tissues

Male C57BL6/J were purchased from the Jackson Laboratory (Sacramento, CA, USA), acclimated in our facility for >1 week and were dosed with 200 mg/kg WIN 18,446 or vehicle (Nutella) alone by oral pipet instillation. Tissues (MLN, liver and testes) were collected 24 h post treatment. Tissues were homogenized in 25 mM Tris buffer (pH 7.4) containing 0.25 M sucrose, 1 mM DTT with metal lysing matrix beads (MP Biomedicals) and centrifuged at 10,000× *g* for 15 min. Supernatants containing cytosolic ALDH1A enzymes (S10) were used to determine RA synthesis capacity as previously reported [39]. Briefly, S10 fractions of liver (15 µg), testis (15 µg) or MLN (30 µg) were incubated with 1 µM retinal and 2 mM NAD for 10 min at 37 °C in an enzyme activity buffer (10 mM HEPES, 150 mM KCl, 2 mM EDTA, pH 7.5). Enzyme activity was quenched with acetonitrile: methanol (1:1, vol: vol) containing all-*trans* RA-d5 (internal standard), and supernatants (3500× *g*, 10 min) were collected to measure RA using LC-MS [39]. RA synthesis capacity (pmol/µg prot/min) was expressed as % of control (animals treated with vehicle).

### 2.6. Statistical Analyses

Student’s T Test or nonparametric Mann–Whitney test was used to compare IBD scores, MLN and PLN by treatment in the steady state study, MLN and LPL by treatment in the inflammation immune cell study, and RA synthesis capacities of control vs. WIN 18,446 treatment. Choice of parametric versus non-parametric statistical tests was determined based on normality tests (Shapiro–Wilk for flow cytometry data and RA synthesis capacity and D’Agostino and Pearson and Shapiro–Wilk for IBD scores and body weight). If data did not pass normality tests, transformation was attempted, and if transformation was unsuccessful then non-parametric methods were used. *p* < 0.05 was considered statistically significant in all analyses.

## 3. Results

### 3.1. WIN 18,446 Significantly Reduces the Severity of Colitis in a Mouse Model of IBD, Smad3^−/−^ Mice

We determined if inhibition of RA synthesis by WIN 18,446 influences colitis in *Smad3*^−/−^ mice. Weights of mice fed WIN 18,446 diet for three weeks prior to the IBD trigger (*H. bilis* infection) did not differ from those fed a control diet, indicating that WIN 18,446 treatment was well tolerated (Figure 1a). However, after infection with *H. bilis*, mice fed the control diet rapidly developed severe clinical signs of illness, with diarrhea starting day 4 post-infection. Four mice fed the control diet were euthanized on day 5 post *H. bilis* infection and five additional mice in the same group were euthanized at day 6 due to severe diarrhea, weight loss, and poor body condition (Figure 1b). In contrast, only two mice fed the diet containing WIN 18,446 showed clinical signs of mild diarrhea (soft/sticky feces) on day 5, but their body condition was good, and their weights were stable. In agreement with decreased clinical signs of IBD in these mice, histologic colitis severity in WIN 18,446 treated mice was significantly reduced at 7 days post *H. bilis* (Figure 1c,d).

### 3.2. Inhibition of RA Synthesis Decreases α4β7^+^ T Cells in Smad3^−/−^ Mice at Steady State

As RA is known to influence expression of gut-homing proteins such as α4β7 integrin on activated T cells, and because treatment with anti-α4β7-specific antibodies can be an effective therapy in human IBDs, we evaluated the effect of RA reduction by ALDH1A inhibition on α4β7 integrin expression in activated T cells in *Smad3*^−/−^ mice at steady state. *Smad3*^−/−^ mice were fed diet with and without WIN 18,446 for 3 weeks and immune cell populations in MLN and PLN were examined. The total cell numbers in MLN, but not PLN, increased with WIN 18,466 treatment. Cell numbers were increased for all the T cell subsets evaluated (Figure 2a,c,e,i; Appendix A) except in the case of CD4^+^ T cells expressing α4β7 integrin where cell numbers remained similar between treatment groups (Figure 2g,k; Appendix A). Percentages of total T cells, total CD4^+^, activated (CD44^Hi^) CD4^+^, and regulatory (FoxP3^+^) CD4^+^ T cells were not significantly different with WIN 18,466 treatment in MLN but the percentage of α4ß7^+^ activated and regulatory CD4^+^ T cells was decreased in MLN (Figure 2h,l; Appendix A). This effect was not observed in activated CD8^+^ T cells (Appendix A). Expression level of α4ß7 integrin was evaluated by determining median fluorescence intensity (MFI) of α4β7 integrin on α4β7^+^ activated and regulatory T cells (Tregs). MFI of α4β7 integrin was significantly decreased (*p* < 0.05) on activated T cells in WIN 18,466 treated mice in both MLN and PLN (MLN mean of MFI value, 2402 ± 82.0 vs. 2096 ± 147; PLN mean of MFI values 1608 ± 25.5 vs. 1505 ± 55.8). MFI of α4β7 integrin was similar between WIN 18,466-treated and control α4β7^+^ Tregs. These data indicate that WIN 18,446 treatment decreases the percent of activated/regulatory CD4^+^ T cells expressing gut-homing receptors and expression level of the integrin on α4β7^+^ CD44^Hi^ CD4^+^ T cells in MLN.

### 3.3. Inhibition of RA Synthesis Alters Multiple Immune Subsets

To determine whether decreased α4ß7 integrin expression on T cells occurs during inflammation in response to RA reduction, we examined T cell populations in MLN and LPL in *H. bilis*-infected *Smad3*^−/−^ mice with and without WIN 18,446 treatment. As mice fed a diet without WIN 18,446 developed severe IBD by day 5 of our first experiment (Figure 1b), we collected tissue samples 4 days after *H. bilis* infection. At this time point, control mice had begun to develop mild diarrhea and upon gross examination had mild thickening of the cecum and colon. Histological analysis conducted with half of the cecum corroborated gross observations (Figure 3a). Immune cells were isolated from the other cecum half, proximal colon and the MLN. Cell numbers of WIN 18,446-treated mice were not significantly changed in either MLN or LPL during inflammation (Appendix A, Cellularity). Similar to steady state (Figure 2), WIN 18,446 treatment decreased α4ß7 expression on activated (CD44^Hi^) CD4^+^ T cells in both MLN and LPL during inflammation (Figure 3b) but not on CD44^+^ CD8^+^ T cells (not shown). Although WIN 18,446 treatment did not result in a change in percent activated CD4^+^ T cells or total numbers of activated CD4^+^ T cells (Appendix A), it altered effector and regulatory T cells expressing the gut-homing integrin (Figure 3c–e, Appendix A). Expression of α4ß7 integrin was reduced on CD4^+^ T cell subsets stained with markers associated with Th17 cells (IL-17F cytokine^+^, Figure 3c or RORγt^+^Foxp3^−^ CD4^+^ T cells, Figure 3d), Tregs (Foxp3^+^ CD4^+^ T cells, not shown) and peripherally induced Tregs (Foxp3^+^RORγt^+^CD4^+^ T cells, Figure 3e). In addition to its influence on gut-homing molecules, RA is known to influence development of Th17 T cells and Tregs in a concentration-dependent manner. To determine if the balance of immunoregulation was shifted following RA reduction, we evaluated effector:Treg ratios by comparing FoxP3^−^:Foxp3^+^ CD4^+^ T cells or by comparing RORγt^+^ FoxP3^−^ effectors: induced Tregs (Figure 3f) in the MLN and LPL of control or WIN 18,446-treated mice. WIN 18,446 treatment was associated with significantly decreased effector: Treg ratios (*p* = 0.0467 and *p* = 0.007, respectively) in MLN.

### 3.4. WIN 18,446 Decreases RA Synthesis in MLN

Previously, we have shown that WIN 18,446 can inhibit human ALDH1A enzymes using purified enzymes [38,39,40]. When mice were treated with this compound, RA levels in testes decreased and long-term treatment resulted in reversible blockage of spermatogenesis [42]. To determine if WIN 18,446 is also effective in inhibiting RA synthesis in the gut draining lymph nodes, we determined the RA synthesis capacity of MLN of mice that were orally treated with either WIN 18,446 or vehicle. As in liver and testes, WIN 18,446 efficiently decreased RA synthesis in MLN (Figure 4), demonstrating that WIN 18,446 given via the oral route is delivered to MLN to inhibit ALDH1A enzymes.

## 4. Discussion

*Smad3*^−/−^ mice are a useful and unique model of IBD. The *SMAD3* locus has been associated with both early onset [43] and adult IBD [44] in genome-wide association studies in IBD patient cohorts, and TGFß signaling has been shown to be deficient in whole tissue and individual cells of IBD patients [32]. *Smad3*^−/−^ mice develop spontaneous IBD if housed in conventional facilities and do not develop IBD in specific pathogen free facilities excluding *Helicobacter*, suggesting that genetic susceptibility itself is not sufficient to trigger the disease. Genome-wide association studies in humans also suggest that the majority of susceptibility genes associated with IBD are not sufficient to induce IBD and environmental factors significantly contribute to the disease development.. Additionally, a significant percentage of infected mice later on develop colon cancer [33], similar to IBD patients that are at increased risk for colon cancer. Therefore, this model recapitulates multiple aspects of human IBD that are distinct from other frequently used IBD models, such as the adoptive transfer models and the chemical injury models, and is a useful tool to decipher mechanisms of development and progression of IBD. Thus, we used this mouse model to determine if RA reduction via ALDH1A inhibition (WIN 18,446 treatment) influences colitis development and severity.

WIN 18,446 strongly inhibits human ALDH1A1 and ALDH1A2, reducing RA in vitro and in vivo [38,39,40]. Long term use of this compound in humans and animals was relatively safe but caused spermatogenesis blockage that can be readily reversed by cessation of the compound [38,42,45]. In addition, we demonstrated that WIN 18,446 can suppress weight gain in a mouse model of diet-induced obesity by inhibiting ALDH1A without significant toxicity after 8 weeks of treatment [39], demonstrating that there is reasonable safety of targeting the RA synthesis pathway for various purposes. Although WIN 18,446 is not currently a viable drug for obesity or male contraception due to its inhibitory effects on ALDH2 which is critical in alcohol metabolism [42], it is a potent tool to decipher the role of RA in health and disease, including IBD. While mouse models are being used to decipher the roles of RA in immune function, these studies are confounded because making mice vitamin A deficient is difficult due to large reserves of vitamin A in the liver. Additionally, the use of non-specific inhibitors for RA-producing enzymes such as citral, a widely used RA synthesis inhibitor in the literature, makes it difficult to interpret results. Although citral can inhibit purified ALDH1A enzymes in vitro, it is not ideal for use in vivo as it is extremely reactive and volatile [46]. In this report, we used WIN 18,446, a proven ALDH1A inhibitor in vivo, to determine whether the RA synthesis pathway can be targeted for IBD treatment and demonstrated that RA reduction can reduce severity of IBD in *Smad3*^−/−^ mice.

Reduced IBD severity in WIN 18,446-treated *Smad3*^−/−^ mice was associated with changes to T cells in the draining lymph nodes, MLN, and gut tissue during inflammation. MLN in *Smad3*^−/−^ mice are enlarged and reactive soon after infection with *H. bilis.* We showed that WIN 18,446 decreased RA synthesis capacity in MLN and altered T cell phenotypes in this tissue early after infection. Thus, we propose that RA reduction alters the early adaptive immune response triggered by *H. bilis*. RA is a natural ligand for retinoic acid receptors (RARs), transcription factors that regulate a diverse array of genes, including α4 and CCR9 [47] that are involved in gut homing. The α4 chain can dimerize with ß7 or ß1 and antibodies against α4, such as natalizumab, are used in IBD treatment. Thus, earlier studies evaluated the role of α4 vs. ß1 in colitis development using the adoptive transfer model of colitis and α4^−/−^ or ß1^−/−^ T cells [48]. These studies showed that α4β7 integrin rather than α4β1 has a more critical role in the ability of T cells to mediate colitis development. We focused on expression of α4β7 integrin in our studies. While there are controversies over what cell types produce the RA needed for upregulation of α4ß7 and CCR9 on T cells, it is generally agreed upon that RA is essential for this process [10,15,17,49,50]. We observed decreased α4β7 integrin expression in MLN and LPL in both Tregs and effector T cells. Our analysis in WIN 18,446-treated *Smad3*^−/−^ mice showed decreased α4β7 integrin in both Tregs (bulk CD4^+^FoxP3^+^ and CD4^+^FoxP3^+^RORγt^+^, which are likely induced during the inflammatory response triggered by *H. bilis* infection [51]) and bulk activated (CD44^Hi^) as well as Th17 effector T cells (CD4^+^FoxP3^−^RORγt^+^). In addition to upregulating gut-homing proteins, RA is also known to alter differentiation of immune cells, nudging differentiation of Th17 cells vs. Treg cells. Also, distinct Treg types have been generated in Vitamin A high and low conditions [30]. Thus, we wondered if the balance of immunoregulation was shifted following WIN 18,446 treatment. To investigate this, we examined the ratio of effector T cells to Treg cells. Though gut homing of both types appears to be affected, ratios of FoxP3^−^CD4^+^: FoxP3^+^Th17: induced Tregs were decreased in WIN 18,446-treated mice, suggesting an overall decreased inflammatory environment during the immune response induced by *H. bilis* infection. Further studies will be needed to determine precise mechanisms by which WIN 18,446-mediated reduction of RA alleviates IBD in *Smad3*^−/−^, including the role of α4β7 integrin and other integrins affected by RA reduction as well as other processes that may be changed due to RA reduction.

## 5. Conclusions

Our data suggest that treatment of *Smad3*^−/−^ mice with WIN 18,446 prior to triggering gut inflammation by *Helicobacter bilis* attenuates IBD in this model. This is associated with a decrease in the gut-homing integrin, α4ß7, on CD4^+^ T cells in the draining lymph nodes, MLN, and colon (LPL). Considering the efficacy of anti-α4ß7 (Vedolizumab^®®^) in IBD, our data suggest that the same pathway may be targeted using small molecule inhibitors that can be taken orally rather than given through i.v. infusion. We are aware of concerns regarding reduction of systemic RA synthesis. However, WIN 18,446 has been used in humans and animals with minimal side effects, and use of targeted drug delivery methods to the colon as well as more specific inhibitors of ALDH1A enzyme (ALDH1A1 vs. ALDH1A2) also can be developed if this pathway is validated for IBD treatment. We propose that RA reduction by inhibiting ALDH1A enzymes is a potential new target for IBD treatment and that additional studies are needed to determine the exact mechanisms by which RA reduction inhibits IBD in this model as well as to determine efficacy of this type of treatment in other models of IBD.

## Figures and Tables

**Figure 1 nutrients-12-02927-f001:**
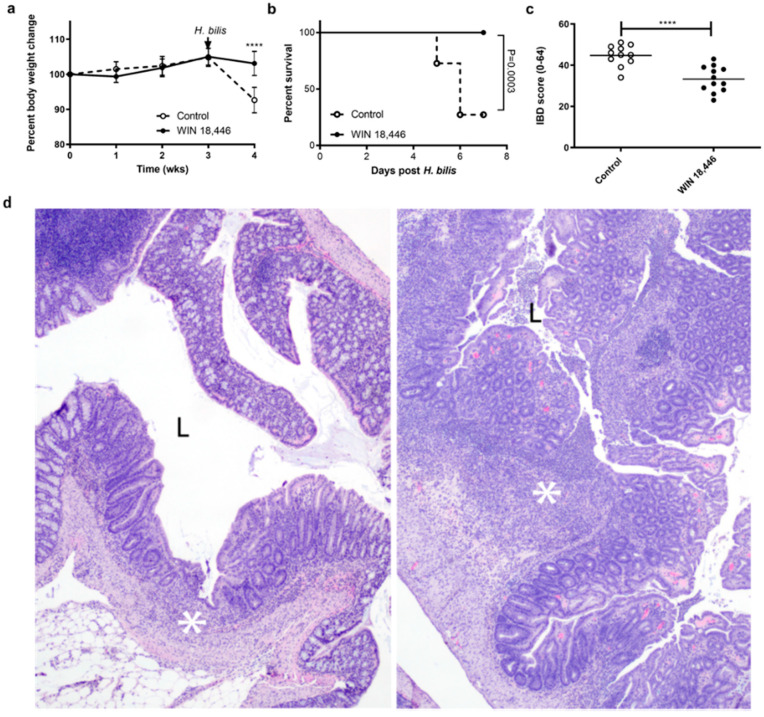
WIN 18,446 treatment protects *Smad3*^−/−^ mice from developing severe colitis. Male and female mice (n = 24; 12/group, age- and sex-matched between the groups) were weighed weekly following the diet initiation. *H. bilis* infection to trigger inflammatory bowel disease (IBD) occurred at 3 weeks post diet initiation (arrow). Mice were necropsied at days 5–7 post *H. bilis* infection. (**a**) Weight change is expressed as percent of baseline. Error bars are SD, **** *p* < 0.0001, Mann–Whitney. (**b**) Survival analysis during the 7 days post *H. bilis* infection was performed using Log-rank test. (**c**) Colitis severity (IBD score) was significantly reduced in *Smad3*^−/−^ mice by treatment with WIN 18,446 as assessed by histological analysis. Horizontal line is the mean score, **** *p* < 0.0001, Student’s T test. (**d**) Representative hematoxylin & eosin (H&E)-stained sections of proximal colon from mice with IBD scores near the mean for each group; left WIN-treated (IBD score 29), right control (IBD score 45). In both colons, there is evidence of ulceration (asterisks). In the control, the ulceration is markedly increased in size and severity; there is active and severe ulceration with cellular exudates in the lumen (L). In contrast, the WIN-treated animal has a healing ulcer where re-epithelization has occurred and there are minimal cells within the lumen (L). Original magnification 100×.

**Figure 2 nutrients-12-02927-f002:**
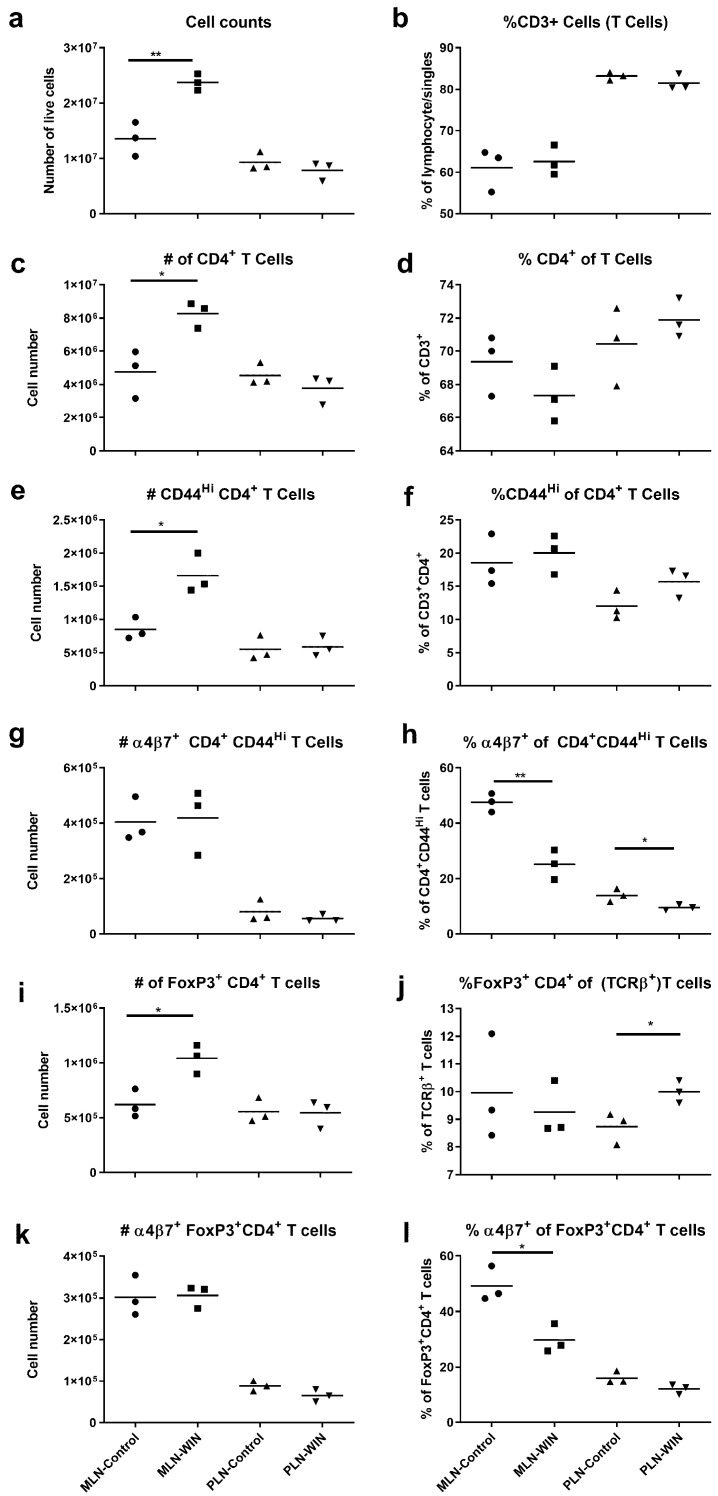
α4β7 integrin expression is decreased in activated CD4^+^ MLN (mesenteric lymph node) T cells of WIN 18,466-treated (WIN) mice compared to controls. Cells from MLN and PLN of mice fed WIN 18,446-containing or control diets (N = 3 mice per group) for 3 weeks were stained with antibodies in two different stains (see also Table 1), (**a**–**h**) Stain 1 and (**i**–**l**) Stain 2, and evaluated using flow cytometry. (**a**) Total live cell counts (hemocytometer) of MLN and PLN preparations. (**b**) Percent CD3^+^ T cells. (**c**) Numbers of CD4^+^ CD3^+^ T cells. (**d**) Percent CD4^+^ of CD3^+^ T cells. (**e**) Numbers of activated (CD44^Hi^) CD4^+^ T cells. (**f**) Percent activated (CD44^Hi^) of CD4^+^ T cells. (**g**) Numbers of α4β7^+^ activated CD4^+^ T cells. (**h**) Percent α4β7^+^ of activated CD4^+^ T cells. (**i**) Numbers of Foxp3^+^ CD4^+^ T (TCRβ^+^) cells. (**j**) Percent Foxp3^+^ CD4^+^ of T cells. (**k**) Numbers of α4β7^+^Foxp3^+^ CD4^+^ T cells. (**l**) Percent α4β7^+^ of Foxp3^+^ CD4^+^ T cells. * *p* < 0.05, ** *p* < 0.01, pair-wise comparisons via Student’s *t*-test or Mann–Whitney test. Note the different origin of y-axes in (**b**,**d**,**j**). Data shown are from one of two independent experiments. The second one is shown in Appendix A.

**Figure 3 nutrients-12-02927-f003:**
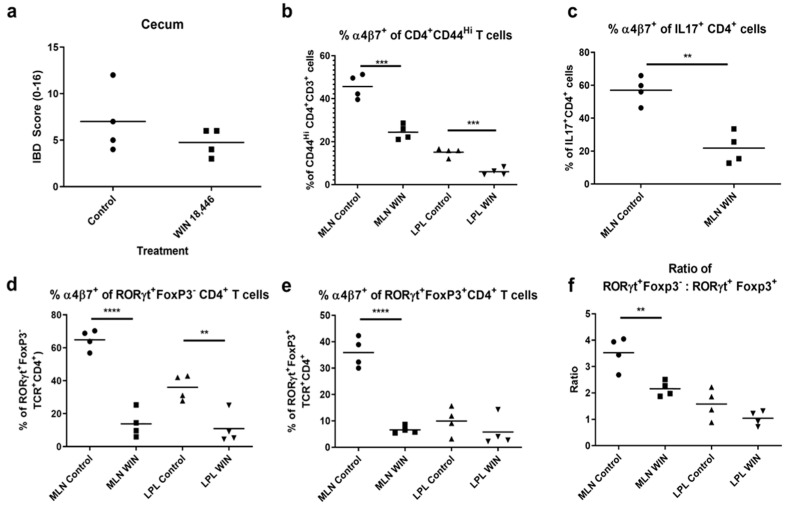
WIN 18,446 treatment results in decreased α4β7 integrin expression in activated (CD44^Hi^) CD4^+^T cells, IL-17^+^ T cells and RORγt^+^CD4^+^ T cells and reduced Th17: induced Treg ratios in MLN and/or LPL during early inflammation. Mice were fed WIN 18,446 (WIN)-containing or control diets (N = 4 mice per group) for 3 weeks and then infected with *H. bilis*. Cell suspensions generated from MLN and LPL were stained with combinations of anti-CD3 or TCRβ, CD4, CD8, CD44, IL-17F, Foxp3, RORγt and α4β7 integrin antibodies in 3 different stains (Stains 1–3, see also Table 1) and evaluated using flow cytometry. (**a**) Histological scores of the cecum (half). Percent α4β7^+^ of (**b**) activated CD4^+^ T cells, (**c**) IL-17^+^CD4^+^ T cells (MLN only), (**d**) RORγt^+^Foxp3^−^CD4^+^ (Th17) T cells, and (**e**) RORγt^+^Foxp3^+^CD4^+^ (induced regulatory) T cells. (**f**) The ratio of Th17 to induced Treg cells. ** *p* < 0.01, *** *p* < 0.001, **** *p* < 0.0001, Student’s *t* test.

**Figure 4 nutrients-12-02927-f004:**
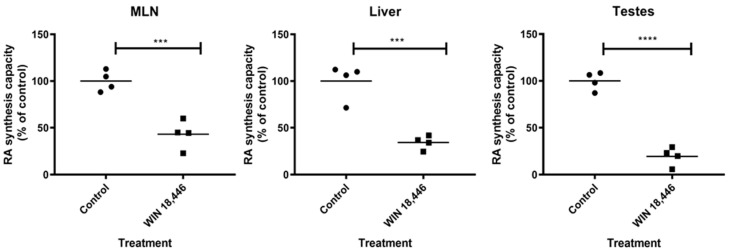
WIN 18,446 significantly suppresses RA synthesis capacity in tissues. S10 fractions containing ALDH1A enzymes were prepared from various tissues collected from mice 24 h following treatment with WIN 18,446 or vehicle alone (Control). RA synthesis capacities of the tissue S10 fractions were determined by measuring their RA production after incubation with substrate, retinal, by LC-MS and expressed as percent of control. Mean is shown (horizontal line). *** *p* < 0.001, **** *p* < 0.0001, Student’s *t* test.

**Table 1 nutrients-12-02927-t001:** Antibody-fluorophore combinations used in two experiments.

	Steady State Experiment/Tissue		Inflammation Experiment/Tissue	
Stains	MLN	PLN	MLN	LPL
Live/Dead	n.d.	n.d.	n.d.	Live/Dead Blue ^3^
Stain 1: Naïve and activated T cells	CD3-APC ^2^ CD4-PerCP-Cy5.5 ^3^ CD8-FITC ^1^ CD44 PE-Cy7 ^1^ α4β7-PE ^3^	same	same	same
Stain 2: Tregs (Foxp3) and Th17 (RORγt)	TCRβ-FITC ^1^ CD4-PerCP-Cy5.5 Foxp3-APC α4β7-PE	TCRβ-FITC CD4-PerCP-Cy5.5 Foxp3-APC α4β7-PE	TCRβ-FITC CD4-PerCP-Cy5.5 Foxp3-APC α4β7-PE RORγt-BV421 ^1^	TCR-β FITC CD4-PerCP-Cy5.5 Foxp3-APC α4β7-PE RORγt-BV421
Stain 3: Intracellular cytokines	CD4-PerCP-Cy5.5 IL-17A-AF647 ^2^ IFNγ-BV421 ^2^ α4β7-PE	same	same	same

Reagent sources: ^1^ BD Biosciences, ^2^ BioLegend, ^3^ eBioscience/Thermofisher. (MLN)—mesenteric lymph node, (PLN)—peripheral lymph nodes, (LPL)—lamina propria leukocytes.

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
