# Peer review of "Protective Effects of ALDH1A Enzyme Inhibition on Helicobacter-Induced Colitis in Smad3−/− Mice are Associated with Altered α4ß7 Integrin Expression on Activated T Cells"

_nutrients, 2020, doi:10.3390/nu12102927_

Round 1
Reviewer 1 Report
The authors appropriately addressed the majority of my concerns. However, they fall short on providing necessary repeats.
While the infection experiment may be sufficiently powered, the phenotyping experiments are not. All dataset should represent at least once repeated experiments. The authors provided a repeat experiment for data shown in Figure 2. It is unclear why the data is not pooled, or why the legend does not state that the data are representative of two experiments now.
For infection experiments and DC characterization, repeats are still lacking.
Three weeks of housing in different cages (presumably the case for differently fed mice) can induce significant cage effects via the microbiota, which may or may not have anything to do with the diet. Given the variance in mucosal immune cell abundances due to this, single experiments with 3 mice per group are not sufficient to allow for conclusions.
Reviewer 2 Report
My previous comments have been addressed and the manuscript is much improved.
Author Response
Reviewer 2 had no further concerns.
Round 2
Reviewer 1 Report
One week does indeed not suffice to repeat the requested experiments.
Nevertheless, single experiments with 3 mice do not yield sufficiently rigid datasets to allow for publication. One solution could be to omit parts of these data in the submitted manuscript, as not all datasets are required for the main message.
Author Response
Please see the attachment.

This manuscript is a resubmission of an earlier submission. The following is a list of the peer review reports and author responses from that submission.
Round 1
Reviewer 1 Report
see attached document

Reviewer 2 Report
This is a very interesting study. Although some of the differences are impressive, I have some comments and questions since some key information is missing.
- What were the typical total cell yields from the mucosal tissue?
- How many live events were typically acquired per sample during the flow cytometry analysis? Much of the data are expressed as percentages, but if the total number of events was low then this could be misleading.
- There is no mention of the types of controls used in the flow cytometry staining (Fc blocking, isotype controls, fluorescence minus one, etc) , and no example dot plots have been provided to validate the specificity of the staining or to exemplify the analysis strategy. These must be provided.
- Information should be provided concerning the IBD scores - how were these data derived? Was there any blinding of the groups to avoid scoring bias?
- Why were some data sets analysed by parametric tests and others using non-parametric tests? Were tests carried out to ascertain whether there was a normal distribution?